# A More Accurate Half-Discrete Multidimensional Hilbert-Type Inequality Involving One Multiple Upper Limit Function

**Yong Hong** [1,2], **Yanru Zhong** [3,*] **and Bicheng Yang** [4]

1    Department of Applied Mathematics, Guangzhou Huashang College, Guangzhou 511300, China
2    College of Mathematics and Statistics, Guangdong University of Finance and Economics,
     Guangzhou 510320, China
3    School of Computer Science and Information Security, Guilin University of Electronic Technology,
     Guilin 541004, China
4    School of Mathematics, Guangdong University of Education, Guangzhou 510303, China
*    Correspondence: 18577399236@163.com

**Abstract:** By means of the weight functions, the idea of introduced parameters, using the transfer formula and Hermite–Hadamard's inequality, a more accurate half-discrete multidimensional Hilbert-type inequality with the homogeneous kernel as $\frac{1}{(x+||k-\xi||_\alpha)^\lambda}$ $(x, \lambda > 0)$ involving one multiple upper limit function is given, which is a new application of Hilbert-type inequalities. The equivalent conditions of the best possible constant factor related to several parameters are considered. The equivalent forms the operator expressions and some particular inequalities are obtained.

**Keywords:** weight function; half-discrete multidimensional Hilbert-type inequality; multiple upper limit function; parameter; beta function; operator expression

**MSC:** 26D15





## 1. Introduction

If $p > 1, \frac{1}{p} + \frac{1}{q} = 1, a_m, b_n \geq 0, 0 < \sum_{m=1}^{\infty} a_m^p < \infty$ and $0 < \sum_{n=1}^{\infty} b_n^q < \infty$, then we have the following discrete Hardy–Hilbert's inequality with the best possible constant factor $\pi / \sin(\frac{\pi}{p})$ (cf. [1], Theorem 315):

$$\sum_{m=1}^{\infty} \sum_{n=1}^{\infty} \frac{a_m b_n}{m+n} < \frac{\pi}{\sin(\pi/p)} \left( \sum_{m=1}^{\infty} a_m^p \right)^{\frac{1}{p}} \left( \sum_{n=1}^{\infty} b_n^q \right)^{\frac{1}{q}}. \tag{1}$$

The integral analogues of (1) named in Hardy–Hilbert's integral inequality was provided as follows (cf. [1], Theorem 316):

$$\int_0^{\infty} \int_0^{\infty} \frac{f(x)g(y)}{x+y} dxdy < \frac{\pi}{\sin(\pi/p)} \left( \int_0^{\infty} f^p(x)dx \right)^{\frac{1}{p}} \left( \int_0^{\infty} g^q(y)dy \right)^{\frac{1}{q}}, \tag{2}$$

with the same best possible factor. The more accurate form of (1) was given as follows (cf. [1], Theorem 323):

$$\sum_{m=1}^{\infty} \sum_{n=1}^{\infty} \frac{a_m b_n}{m+n-1} < \frac{\pi}{\sin(\pi/p)} \left( \sum_{m=1}^{\infty} a_m^p \right)^{\frac{1}{p}} \left( \sum_{n=1}^{\infty} b_n^q \right)^{\frac{1}{q}}. \tag{3}$$

In Equations (1)–(3), with their extensions, played an important role in analysis and its applications (cf. [2–15]).

The following half-discrete Hilbert-type inequality was provided in 1934 (cf. [1], Theorem 351): If $K(x)\,(x > 0)$ is decreasing, $p > 1$, $\frac{1}{p} + \frac{1}{q} = 1$, $0 < \varphi(s) = \int_0^\infty K(x)x^{s-1}dx < \infty$, $f(x) \geq 0$, $0 < \int_0^\infty f^p(x)dx < \infty$, then

$$\sum_{n=1}^\infty n^{p-2}\left(\int_0^\infty K(nx)f(x)dx\right)^p < \varphi^p\left(\frac{1}{q}\right)\int_0^\infty f^p(x)dx. \tag{4}$$

Some new extensions of (3) were given by [16–19].

In 2006, using the Euler–Maclaurin summation formula, Krnic et al. [20] gave an extension of (1) with the kernel as $\frac{1}{(m+n)^\lambda}\,(0 < \lambda \leq 4)$. In 2019–2020, following the results of [20], Adiyasuren et al. [21] provided an extension of (1) involving partial sums, and Mo et al. [22] gave an extension of (2) involving the upper limit functions. In 2016–2017, Hong et al. [23,24] considered some equivalent statements of the extensions of (1) and (2) with a few parameters. Some further results were provided by [25–27].

In this paper, we extend Mo's work in [22] to half-discrete multidimensional Hilbert-type inequalities. By means of the weight functions and the idea of introduced parameters, using the transfer formula and Hermite–Hadamard's inequality, a more accurate half-discrete multidimensional Hilbert-type inequality with the homogeneous kernel as $\frac{1}{(x+||k-\xi||_\alpha)^\lambda}\,(x, \lambda > 0, \xi \in [0, \frac{1}{2}])$, involving one multiple upper limit function and the beta function, is given. The equivalent conditions of the best possible constant factor related to several parameters are provided. The equivalent forms, the operator expressions and some particular inequalities are obtained. Our main results are new applications of Hilbert-type inequalities involving multiple upper limit functions.

## 2. Some Formulas and Preserving Lemmas

Hereinafter in this paper, we assume that $p > 1$, $\frac{1}{p} + \frac{1}{q} = 1$, $\lambda > 0$, $\lambda_1, \lambda_2 \in (0, \lambda)$, $m, n \in \mathrm{N} = \{1, 2, \cdots\}$, $\alpha \in (0, 1]$, $\xi \in [0, \frac{1}{2}]$, $\hat{\lambda}_1 := \frac{\lambda - \lambda_2}{p} + \frac{\lambda_1}{q}$, $\hat{\lambda}_2 := \frac{\lambda - \lambda_1}{q} + \frac{\lambda_2}{p}$,

$$||y||_\alpha := \left(\sum_{i=1}^n |y_i|^\alpha\right)^{\frac{1}{\alpha}} \quad (y = (y_1, \cdots, y_n) \in \mathrm{R}^n).$$

For $f(x) := F_0(x) \geq 0$, define the following multiple upper limit functions $F_i(x) := \int_0^x F_{i-1}(t)dt\,(x \geq 0)$, inductively, satisfying $F_i(0) = 0$, and

$$F_i(x) = o(e^{tx})\,(t > 0, i = 1, \cdots, m; x \to \infty),$$

which means that for $t > 0$, $\frac{F_i(x)}{e^{tx}} \to 0\,(x \to \infty)$. We also assume that $F_m(x)$, $a_k = (a_{k_1}, \cdots, a_{k_n}) \geq 0\,(x \in \mathrm{R}_+ = (0, \infty), k = (k_1, \cdots, k_n) \in \mathrm{N}^n)$, such that

$$0 < \int_0^\infty x^{p(1-m-\hat{\lambda}_1)-1}F_m^p(x)dx < \infty \quad and \quad 0 < \sum_k ||k - \xi||_\alpha^{q(n-\hat{\lambda}_2)-n}a_k^q < \infty.$$

For $M > 0$, $\psi(u)\,(u > 0)$ is a nonnegative measurable function; we have the following transfer formula (cf. [3], (9.3.3)):

$$\int \cdots \int_{\{y \in \mathrm{R}_+^n; 0 < \sum_{i=1}^n (\frac{y_i}{M})^\alpha \leq 1\}} \psi\left(\sum_{i=1}^n \left(\frac{y_i}{M}\right)^\alpha\right)dy_1 \cdots dy_n = \frac{M^n\Gamma(\frac{1}{\alpha})}{\alpha^n\Gamma(\frac{n}{\alpha})}\int_0^1 \psi(u)u^{\frac{n}{\alpha}-1}du. \tag{5}$$

In particular, (i) in view of $||y||_\alpha = M[\sum_{i=1}^n (\frac{y_i}{M})^\alpha]^{\frac{1}{\alpha}}$, by (5), we have

$$
\begin{aligned}
\int_{\mathrm{R}_+^n} \varphi(||y||_\alpha)dy &= \lim_{M \to \infty} \int \cdots \int_{\{y \in \mathrm{R}_+^n; 0 < \sum_{i=1}^n (\frac{y_i}{M})^\alpha \leq 1\}} \varphi\left(M\left[\sum_{i=1}^n \left(\frac{y_i}{M}\right)^\alpha\right]^{\frac{1}{\alpha}}\right)dy_1 \cdots dy_n \\
&= \lim_{M \to \infty} \frac{M^n\Gamma(\frac{1}{\alpha})}{\alpha^n\Gamma(\frac{n}{\alpha})}\int_0^1 \varphi(Mu^{\frac{1}{\alpha}})u^{\frac{n}{\alpha}-1}du \overset{v=Mu^{\frac{1}{\alpha}}}{=\!=\!=} \frac{\Gamma(\frac{1}{\alpha})}{\alpha^{n-1}\Gamma(\frac{n}{\alpha})}\int_0^\infty \varphi(v)v^{n-1}dv;
\end{aligned} \tag{6}
$$

(ii) for $\psi(u) = \varphi(Mu^{\frac{1}{\alpha}}) = 0.u < \frac{b^\alpha}{M^\alpha}$ $(b > 0)$, by (5), we have

$$\int_{\{y \in R^n_+, ||y||_\alpha \geq b\}} \varphi(||y||_\alpha) dy = \lim_{M \to \infty} \frac{M^n \Gamma(\frac{1}{\alpha})}{\alpha^n \Gamma(\frac{n}{\alpha})} \int_{\frac{b^\alpha}{M^\alpha}}^1 \varphi(Mu^{\frac{1}{\alpha}}) u^{\frac{n}{\alpha}-1} du = \frac{\Gamma(\frac{1}{\alpha})}{\alpha^{n-1}\Gamma(\frac{n}{\alpha})} \int_b^\infty \varphi(v) v^{n-1} dv. \tag{7}$$

**Lemma 1.** *For $s > 0, \alpha \in (0,1], \xi \in [0, \frac{1}{2}], A_\xi := \{y = \{y_1, \cdots, y_n\}; y_i > \xi (i = 1, \cdots, n)\}$,*

*define the following function:*

$$g_x(y) := \frac{1}{(x + ||y - \xi||_\alpha)^s} = \frac{1}{\{x + [\sum_{i=1}^n (y_i - \xi)^\alpha]^{1/\alpha}\}^s} (x > 0, y = (y_1, \cdots, y_n) \in A_\xi).$$

*Then we have $\frac{\partial}{\partial y_j} g_x(y) < 0, \frac{\partial^2}{\partial y_j^2} g_x(y) > 0 (y \in A_\xi; j = 1, \cdots, n)$.*

**Proof.** We obtain that for $s > 0, \alpha \in (0,1], \xi \in [0, \frac{1}{2}], y \in A_\xi$,

$$\frac{\partial}{\partial y_j} g_x(y) = \frac{-s[\sum_{i=1}^n (y_i - \xi)^\alpha]^{\frac{1}{\alpha}-1}(y_j - \xi)^{\alpha-1}}{\{x + [\sum_{i=1}^n (y_i - \xi)^\alpha]^{1/\alpha}\}^{s+1}} < 0,$$

$$\frac{\partial^2}{\partial y_j^2} g_x(y) = \frac{s(s+1)[\sum_{i=1}^n (y_i - \xi)^\alpha]^{\frac{2}{\alpha}-2}(y_j - \xi)^{2\alpha-2}}{\{x + [\sum_{i=1}^n (y_i - \xi)^\alpha]^{1/\alpha}\}^{s+2}}$$

$$+ \frac{s(1-\alpha)[\sum_{i=1}^n (y_i - \xi)^\alpha]^{\frac{1}{\alpha}-2}(y_j - \xi)^{\alpha-2}}{\{x + [\sum_{i=1}^n (y_i - \xi)^\alpha]^{1/\alpha}\}^{s+1}} \Big[\sum_{i=1}^n (y_i - \xi)^\alpha - (y_j - \xi)^\alpha\Big] > 0.$$

The lemma is proved. □

Note. In the same way, for $s_2 \leq n, \alpha \in (0,1], \xi \in [0, \frac{1}{2}], y \in A_\xi$, we can find that

$$\frac{\partial}{\partial y_j} ||y - \xi||_\alpha^{s_2-n} \leq 0, \frac{\partial^2}{\partial y_j^2} ||y - \xi||_\alpha^{s_2-n} \geq 0 (j = 1, \cdots, n), \tag{8}$$

and then for $s_2 \leq n, \alpha \in (0,1], \xi \in [0, \frac{1}{2}], h_x(y) := g_x(y)||y - \xi||_\alpha^{s_2-n} (x > 0, y \in A_\xi)$, by Lemma 1, we have

$$\frac{\partial}{\partial y_j} h_x(y) = ||y - \xi||_\alpha^{s_2-n} \frac{\partial}{\partial y_j} g_x(y) + g_x(y) \frac{\partial}{\partial y_j} ||y - \xi||_\alpha^{s_2-n} < 0, \frac{\partial^2}{\partial y_j^2} h_x(y) =$$

$$\frac{\partial}{\partial y_j} ||y - \xi||_\alpha^{s_2-n} \frac{\partial}{\partial y_j} g_x(y) + ||y - \xi||_\alpha^{s_2-n} \frac{\partial^2}{\partial y_j^2} g_x(y) + \frac{\partial}{\partial y_j} g_x(y) \frac{\partial}{\partial y_j} ||y - \xi||_\alpha^{s_2-n} \tag{9}$$

$$+ g_x(y) \frac{\partial^2}{\partial y^2} ||y - \xi||_\alpha^{s_2-n} > 0, (j = 1, \cdots, n).$$

**Lemma 2.** *For $c > 0$, we have the following inequalities:*

$$\frac{\Gamma(\frac{1}{\alpha})}{c\alpha^{n-1}\Gamma(\frac{n}{\alpha})} < \sum_k ||k||_\alpha^{-c-n} < \frac{2^c \Gamma(\frac{1}{\alpha})}{c\alpha^{n-1}\Gamma(\frac{n}{\alpha})}, \tag{10}$$

*where $\sum_k G(k) = \sum_{k_n=1}^\infty \cdots \sum_{k_1=1}^\infty G(k_1, \cdots, k_n)$ $(G(k)(\geq 0)$ is the term of multiple series with respect to $k \in N^n$).*

**Proof.** By (8) (for $\xi = 0$), in view of $-c - n < 0$, we find that

$$\frac{\partial}{\partial y_j} ||y||_\alpha^{-c-n} < 0, \frac{\partial^2}{\partial y_j^2} ||y||_\alpha^{-c-n} > 0 (j = 1, \cdots, n),$$

and then by Hermite–Hadamard's inequality (cf. [28]) and (7), we have

$$\sum_k ||k||_\alpha^{-c-n} < \int_{\{y\in R_+^n, ||y||_\alpha \geq \frac{1}{2}\}} ||y||_\alpha^{-c-n} dy = \frac{\Gamma(\frac{1}{\alpha})}{\alpha^{n-1}\Gamma(\frac{n}{\alpha})} \int_{\frac{1}{2}}^\infty v^{-c-n} v^{n-1} dv = \frac{2^c \Gamma(\frac{1}{\alpha})}{c\alpha^{n-1}\Gamma(\frac{n}{\alpha})}.$$

By the decreasingness property of series and (7), it follows that

$$\sum_k ||k||_\alpha^{-c-n} > \int_{\{y\in R_+^n, ||y||_\alpha \geq 1\}} ||y||_\alpha^{-c-n} dy = \frac{\Gamma(\frac{1}{\alpha})}{\alpha^{n-1}\Gamma(\frac{n}{\alpha})} \int_1^\infty v^{-c-1} dv = \frac{\Gamma(\frac{1}{\alpha})}{c\alpha^{n-1}\Gamma(\frac{n}{\alpha})},$$

namely, inequalities (10) follow.

The lemma is proved. □

**Lemma 3.** *For $s > 0$, we define the following weight functions:*

$$\varpi_s(s_2, x) := x^{s-s_2} \sum_k \frac{||k - \xi||_\alpha^{s_2-n}}{(x + ||k - \xi||_\alpha)^s} \, (x \in R_+), \tag{11}$$

$$\omega_s(s_1, k) := ||k - \xi||_\alpha^{s-s_1} \int_0^\infty \frac{x^{s_1-1}}{(x + ||k - \xi||_\alpha)^s} dx \, (k \in N^n), \tag{12}$$

(i) *for $0 < s_2 < s, s_2 \leq n$, we have the following inequalities:*

$$\frac{\Gamma(\frac{1}{\alpha})}{\alpha^{n-1}\Gamma(\frac{n}{\alpha})} B(s_2, s - s_2)(1 - \theta_s(s_2, x)) < \varpi_s(s_2, x) < \frac{\Gamma(\frac{1}{\alpha})}{\alpha^{n-1}\Gamma(\frac{n}{\alpha})} B(s_2, s - s_2)(x \in R_+), \tag{13}$$

*where,*

$$\theta_s(s_2, x) := \frac{1}{B(s_2, s - s_2)} \int_0^{1/x} \frac{u^{s_2-1}}{(1 + u)^s} du = O(\frac{1}{x^{s_2}}) \in (0, 1),$$

*which means that $x^{s_2}\theta_s(s_2, x)$ is bounded for $x \in R_+$. and*

$$B(u, v) := \int_0^\infty \frac{t^{u-1}}{(1 + t)^{u+v}} dt \, (u, v > 0)$$

*is the beta function.*

(ii) *for $0 < s_1 < s$, we have the following expression:*

$$\omega_s(s_1, k) = B(s_1, s - s_1) (y \in R_+^n). \tag{14}$$

**Proof.** (i) For $0 < s_2 < s, s_2 \leq n$, by (9), (11) and Hermite–Hadamard's inequality (cf. [28]), we have

$$\varpi_s(s_2, x) < x^{s-s_2} \int_{A_{1/2}} \frac{||y-\xi||_\alpha^{s_2-n}}{(x+||y-\xi||_\alpha)^s} dy$$

$$\leq x^{s-s_2} \int_{A_\xi} \frac{||y-\xi||_\alpha^{s_2-n}}{(x+||y-\xi||_\alpha)^s} dy = x^{s-s_2} \int_{R_+^n} \frac{||u||_\alpha^{s_2-n}}{(x+||u||_\alpha)^s} du.$$

Setting $\varphi(v) := \frac{v^{s_2-n}}{(x+v)^s}$, by (6), it follows that

$$\varpi_s(s_2, x) < x^{s-s_2} \int_{R_+^n} \varphi(||u||_\alpha) du = x^{s-s_2} \frac{\Gamma(\frac{1}{\alpha})}{\alpha^{n-1}\Gamma(\frac{n}{\alpha})} \int_0^\infty \varphi(v) v^{n-1} dv$$

$$= x^{s-s_2} \frac{\Gamma(\frac{1}{\alpha})}{\alpha^{n-1}\Gamma(\frac{n}{\alpha})} \int_0^\infty \frac{v^{s_2-1}}{(x+v)^s} dv \overset{t=v/x}{=} \frac{\Gamma(\frac{1}{\alpha})}{\alpha^{n-1}\Gamma(\frac{n}{\alpha})} \int_0^\infty \frac{t^{s_2-1}}{(1+t)^s} dt$$

$$= \frac{\Gamma(\frac{1}{\alpha})}{\alpha^{n-1}\Gamma(\frac{n}{\alpha})} B(s_2, s - s_2).$$

In view of the decreasingness property of series, we find

$$
\begin{aligned}
\omega_s(s_2, x) &> x^{s-s_2} \int_{\{y \in R^n_+; \|y\|_\alpha \geq 1\}} \varphi(\|y\|_\alpha) dy = x^{s-s_2} \frac{\Gamma(\frac{1}{\alpha})}{\alpha^{n-1}\Gamma(\frac{n}{\alpha})} \int_1^\infty \varphi(v) v^{n-1} dv \\
&= x^{s-s_2} \frac{\Gamma(\frac{1}{\alpha})}{\alpha^{n-1}\Gamma(\frac{n}{\alpha})} \int_1^\infty \frac{v^{s_2-1}}{(x+v)^s} dv \overset{u=v/x}{=} \frac{\Gamma(\frac{1}{\alpha})}{\alpha^{n-1}\Gamma(\frac{n}{\alpha})} \int_{1/x}^\infty \frac{u^{s_2-1}}{(1+u)^s} du \\
&= \frac{\Gamma(\frac{1}{\alpha})}{\alpha^{n-1}\Gamma(\frac{n}{\alpha})} B(s_2, s-s_2)(1 - \theta_s(s_2, x)) > 0, \\
0 &< \theta_s(s_2, x) = \frac{1}{B(s_2, s-s_2)} \int_0^{1/x} \frac{u^{s_2-1}}{(1+u)^s} du \\
&\leq \frac{1}{B(s_2, s-s_2)} \int_0^{1/x} u^{s_2-1} du = \frac{1}{s_2 B(s_2, s-s_2)} \frac{1}{x^{s_2}} \quad (x \in R_+).
\end{aligned}
$$

Hence, we have (13).

(ii) Setting $u = \frac{x}{\|k-\xi\|_\alpha}$ in (12), we find

$$
\omega_s(s_1, k) = \|k - \xi\|_\alpha^{s-s_1} \int_0^\infty \frac{(u\|k-\xi\|_\alpha)^{s_1-1} \|k-\xi\|_\alpha}{(u\|k-\xi\|_\alpha + \|k-\xi\|_\alpha)^s} du = \int_0^\infty \frac{u^{s_1-1}}{(u+1)^s} du = B(s_1, s-s_1),
$$

and then (14) follows.

The lemma is proved. $\square$

We indicate the following gamma function (cf. [29]): $\Gamma(\alpha) := \int_0^\infty e^{-t} t^{\alpha-1} dt \, (\alpha > 0)$, satisfying $\Gamma(\alpha+1) = \alpha\Gamma(\alpha)(\alpha > 0)$ and $B(u, v) = \frac{1}{\Gamma(u+v)}\Gamma(u)\Gamma(v)(u, v > 0)$. By the definition of the gamma function, for $\lambda, x > 0$, the following expression holds:

$$
\frac{1}{(x + \|k-\xi\|_\alpha)^{\lambda+m}} = \frac{1}{\Gamma(\lambda+m)} \int_0^\infty t^{\lambda+m-1} e^{-(x+\|k-\xi\|_\alpha)t} dt. \tag{15}
$$

**Lemma 4.** *For $t > 0$, we have the following expression:*

$$
\int_0^\infty e^{-tx} f(x) dx = t^m \int_0^\infty e^{-tx} F_m(x) dx. \tag{16}
$$

**Proof.** Since $F_1(0) = 0$, $F_1(x) = o(e^{tx}) \, (t > 0; x \to \infty)$, for $m = 1$, we find

$$
\begin{aligned}
\int_0^\infty e^{-tx} f(x) dx &= \int_0^\infty e^{-tx} dF_1(x) = e^{-tx} F_1(x)|_0^\infty - \int_0^\infty F_1(x) de^{-tx} \\
&= \lim_{x \to \infty} \frac{F_1(x)}{e^{tx}} + t \int_0^\infty e^{-tx} F_1(x) dx = t \int_0^\infty e^{-tx} F_1(x) dx.
\end{aligned}
$$

Hence, (16) follows. Assuming that for $m = i$, (16) is valid, then for $m = i+1$, since $F_{i+1}(0) = 0$, $F_{i+1}(x) = o(e^{tx}) \, (t > 0, x \to \infty)$, we have

$$
\int_0^\infty e^{-tx} F_i(x) dx = t \int_0^\infty e^{-tx} F_{i+1}(x) dx,
$$

and then

$$
\int_0^\infty e^{-tx} f(x) dx = t^i \int_0^\infty e^{-tx} F_i(x) dx = t^{i+1} \int_0^\infty e^{-tx} F_{i+1}(x) dx.
$$

By mathematical induction, expression (16) follows for $m \in N$.

The lemma is proved. $\square$

**Lemma 5.** *We have the following inequality:*

$$
\begin{aligned}
I_{\lambda+m} := \sum_k \int_0^\infty \frac{F_m(x) a_k}{(x + \|k-\xi\|_\alpha)^{\lambda+m}} dx &< \left( \frac{\Gamma(\frac{1}{\alpha})}{\alpha^{n-1}\Gamma(\frac{n}{\alpha})} B(\lambda_2, \lambda+m-\lambda_2) \right)^{\frac{1}{p}} B^{\frac{1}{q}}(\lambda_1+m, \lambda-\lambda_1) \\
&\times \left[ \int_0^\infty x^{p(1-m-\hat{\lambda}_1)-1} F_m^p(x) dx \right]^{\frac{1}{p}} \left[ \sum_k \|k-\xi\|_\alpha^{q(n-\hat{\lambda}_2)-n} a_k^q \right]^{\frac{1}{q}}.
\end{aligned} \tag{17}
$$

**Proof.** By Hölder's inequality (cf. [28]), and Lebesgue term by term integral theorem (cf. [30]), we obtain

$$
\begin{aligned}
I_{\lambda+m} &= \sum_k \int_0^\infty \frac{1}{(x+||k-\xi||_\alpha)^{\lambda+m}} \Big[ \frac{||k-\xi||_\alpha^{(\lambda_2-n)/p}}{x^{(\lambda_1+m-1)/q}} F_m(x) \Big] \Big[ \frac{x^{(\lambda_1+m-1)/q}}{||k-\xi||_\alpha^{(\lambda_2-n)/p}} a_k \Big] dx \\
&\le \left\{ \int_0^\infty \Big[ \sum_k \frac{1}{(x+||k-\xi||_\alpha)^{\lambda+m}} \frac{||k-\xi||_\alpha^{\lambda_2-n}}{x^{(\lambda_1+m-1)(p-1)}} \Big] F_m^p(x) dx \right\}^{\frac{1}{p}} \\
&\quad \times \left\{ \sum_k \Big[ \int_0^\infty \frac{1}{(x+||k-\xi||_\alpha)^{\lambda+m}} \frac{x^{\lambda_1+m-1}}{||k-\xi||_\alpha^{(\lambda_2-n)(q-1)}} dx \Big] a_k^q \right\}^{\frac{1}{q}} \\
&= \Big[ \int_0^\infty \varpi_{\lambda+m}(\lambda_2, x) x^{p(1-m-\hat\lambda_1)-1} F_m^p(x) dx \Big]^{\frac{1}{p}} \\
&\quad \times \Big[ \sum_k \omega_{\lambda+m}(\lambda_1+m, k) ||k-\xi||_\alpha^{q(n-\hat\lambda_2)-n} a_k^q \Big]^{\frac{1}{q}}.
\end{aligned}
$$

Therefore, by (13) and (14) (for $s = \lambda + m, s_1 = \lambda_1 + m, s_2 = \lambda_2$), we have (17). The lemma is proved. □

## 3. Main Results

**Theorem 1.** *We have the following more accurate half-discrete multidimensional Hilbert-type inequality involving one multiple supper limit function:*

$$
\begin{aligned}
I := \sum_k \int_0^\infty \frac{f(x)a_k}{(x+||k-\xi||_\alpha)^\lambda} dx &< \prod_{i=0}^{m-1} (\lambda+i) \Big( \frac{\Gamma(\frac{1}{\alpha})}{\alpha^{n-1}\Gamma(\frac{n}{\alpha})} B(\lambda_2, \lambda+m-\lambda_2) \Big)^{\frac{1}{p}} B^{\frac{1}{q}}(\lambda_1+m, \lambda-\lambda_1) \\
&\times \Big[ \int_0^\infty x^{p(1-m-\hat\lambda_1)-1} F_m^p(x) dx \Big]^{\frac{1}{p}} \Big[ \sum_k ||k-\xi||_\alpha^{q(n-\hat\lambda_2)-n} a_k^q \Big]^{\frac{1}{q}}.
\end{aligned}
\tag{18}
$$

In particular, for $\lambda_1 + \lambda_2 = \lambda$, we reduce (18) to the following:

$$
\begin{aligned}
I = \sum_k \int_0^\infty \frac{f(x)a_k}{(x+||k-\xi||_\alpha)^\lambda} dx &< \Big( \frac{\Gamma(\frac{1}{\alpha})}{\alpha^{n-1}\Gamma(\frac{n}{\alpha})} \Big)^{\frac{1}{p}} \prod_{i=0}^{m-1} (\lambda_1+i) B(\lambda_1, \lambda_2) \\
&\times \Big[ \int_0^\infty x^{p(1-m-\lambda_1)-1} F_m^p(x) dx \Big]^{\frac{1}{p}} \Big[ \sum_k ||k-\xi||_\alpha^{q(n-\lambda_2)-n} a_k^q \Big]^{\frac{1}{q}}.
\end{aligned}
\tag{19}
$$

where the constant factor $\Big( \frac{\Gamma(\frac{1}{\alpha})}{\alpha^{n-1}\Gamma(\frac{n}{\alpha})} \Big)^{\frac{1}{p}} \prod_{i=0}^{m-1} (\lambda_1+i) B(\lambda_1, \lambda_2)$ is the best possible.

**Proof.** Using (15) and (16), in view of Lebesgue term by term integral theorem (cf. [30]), we find

$$
\begin{aligned}
I &= \frac{1}{\Gamma(\lambda)} \sum_k \int_0^\infty f(x) a_k \Big[ \int_0^\infty t^{\lambda-1} e^{-(x+||k-\xi||_\alpha)t} dt \Big] dx \\
&= \frac{1}{\Gamma(\lambda)} \int_0^\infty t^{\lambda-1} \Big( \int_0^\infty e^{-xt} f(x) dx \Big) \Big( \sum_k e^{-||k-\xi||_\alpha t} a_k \Big) dt \\
&= \frac{1}{\Gamma(\lambda)} \int_0^\infty t^{\lambda-1} \Big( t^m \int_0^\infty e^{-xt} F_m(x) dx \Big) \Big( \sum_k e^{-||k-\xi||_\alpha t} a_k \Big) dt \\
&= \frac{1}{\Gamma(\lambda)} \sum_k \int_0^\infty F_m(x) a_k \Big[ \int_0^\infty t^{\lambda+m-1} e^{-(x+||k-\xi||_\alpha)t} dt \Big] dx \\
&= \frac{\Gamma(\lambda+m)}{\Gamma(\lambda)} \sum_k \int_0^\infty \frac{F_m(x) a_k}{(x+||k-\varsigma||_\alpha)^{\lambda+m}} dx = \prod_{i=0}^{m-1} (\lambda+i) I_{\lambda+m}.
\end{aligned}
$$

Then by (17), we have (18).

For $\lambda_1 + \lambda_2 = \lambda$ in (18), we have (19). For any $0 < \varepsilon < p\lambda_1$, we set

$$
\tilde{f}(x) := \begin{cases} 0, & 0 < x < 1, \\ x^{\lambda_1+m-\frac{\varepsilon}{p}-1}, & x \ge 1 \end{cases}, \quad \tilde{a}_k := ||k||_\alpha^{\lambda_2-\frac{\varepsilon}{q}-n} \ (k \in N^n).
$$

We obtain that for $0 < x < 1, \widetilde{F}_1(x) = 0$; for $x \geq 1$,

$$\widetilde{F}_1(x) = \int_1^x \widetilde{f}(t)dt \leq \int_0^x t^{\lambda_1 - \frac{\varepsilon}{p} - 1}dt = \frac{1}{\lambda_1 - \frac{\varepsilon}{p}}x^{\lambda_1 - \frac{\varepsilon}{p}}.$$

Inductively, we find that $\widetilde{F}_i(x) = o(e^{tx})\,(t > 0, x \to \infty)$ and

$$\widetilde{F}_i(x) = 0, 0 \leq x < 1; \widetilde{F}_i(x) \leq \frac{1}{\prod_{j=0}^{i-1}(\lambda_1 + j - \frac{\varepsilon}{p})}x^{\lambda_1 + i - \frac{\varepsilon}{p} - 1}\,(x \geq 1; i = 1. \cdots, m).$$

If there exists a positive constant $M(\leq (\frac{\Gamma(\frac{1}{\alpha})}{\alpha^{n-1}\Gamma(\frac{n}{\alpha})})^{\frac{1}{p}}\prod_{i=0}^{m-1}(\lambda_1 + i)B(\lambda_1, \lambda_2))$, such that

(19) is valid when we replace $(\frac{\Gamma(\frac{1}{\alpha})}{\alpha^{n-1}\Gamma(\frac{n}{\alpha})})^{\frac{1}{p}}\prod_{i=0}^{m-1}(\lambda_1 + i)B(\lambda_1, \lambda_2)$ by $M$, then in particular, for $\xi = 0$, we still have

$$\widetilde{I} := \sum_k \int_0^\infty \frac{\widetilde{f}(x)\widetilde{a}_k}{(x + ||k||_\alpha)^{\lambda + m}}dx < M[\int_0^\infty x^{p(1-m-\lambda_1)-1}\widetilde{F}_m^p(x)dx]^{\frac{1}{p}}[\sum_k ||k||_\alpha^{q(n-\lambda_2)-n}\widetilde{a}_k^q]^{\frac{1}{q}}. \tag{20}$$

By (10), we obtain

$$\widetilde{J} := [\int_0^\infty x^{p(1-m-\lambda_1)-1}\widetilde{F}_m^p(x)dx]^{\frac{1}{p}}[\sum_k ||k||_\alpha^{q(n-\lambda_2)-n}\widetilde{a}_k^q]^{\frac{1}{q}}$$
$$< [\prod_{i=0}^{m-1}(\lambda_1 + i - \frac{\varepsilon}{p})]^{-1}(\int_1^\infty x^{-\varepsilon-1}dx)^{\frac{1}{p}}(\sum_k ||k||_\alpha^{-\varepsilon-n})^{\frac{1}{q}} \tag{21}$$
$$= \frac{1}{\varepsilon}[\prod_{i=0}^{m-1}(\lambda_1 + i - \frac{\varepsilon}{p})]^{-1}(\frac{2^\varepsilon \Gamma(\frac{1}{\alpha})}{\alpha^{n-1}\Gamma(\frac{n}{\alpha})})^{\frac{1}{q}}.$$

By (10), we also find that $\frac{1}{c-\varepsilon}\sum_k ||k||_\alpha^{-c-n} = O(1)\,(c = \lambda_1 + \frac{\varepsilon}{q})$, where $O(1)$ is bounded for any $\varepsilon > 0$. For $s = \lambda > 0, s_1 = \lambda_1 - \frac{\varepsilon}{p} \in (0, s)$ in (12) and (14), by (10), we obtain

$$\widetilde{I} := \sum_k ||k||_\alpha^{-\varepsilon-n}[||k||_\alpha^{(\lambda_2 + \frac{\varepsilon}{p})}\int_1^\infty \frac{x^{(\lambda_1 - \frac{\varepsilon}{p})-1}}{(x + ||k||_\alpha)^\lambda}dx]$$
$$= \sum_k ||k||_\alpha^{-\varepsilon-n}[||k||_\alpha^{(\lambda_2 + \frac{\varepsilon}{p})}\int_0^\infty \frac{x^{(\lambda_1 - \frac{\varepsilon}{p})-1}}{(x + ||k||_\alpha)^\lambda}dx - ||k||_\alpha^{(\lambda_2 + \frac{\varepsilon}{p})}\int_0^1 \frac{x^{(\lambda_1 - \frac{\varepsilon}{p})-1}}{(x + ||k||_\alpha)^\lambda}dx]$$
$$\geq \sum_k ||k||_\alpha^{-\varepsilon-n}[\omega_\lambda(\lambda_1 - \frac{\varepsilon}{p}, k) - ||k||_\alpha^{(\lambda_2 + \frac{\varepsilon}{p})}\int_0^1 \frac{x^{(\lambda_1 - \frac{\varepsilon}{p})-1}}{||k||_\alpha^\lambda}dx]$$
$$= \sum_k ||k||_\alpha^{-\varepsilon-n}\omega_\lambda(\lambda_1 - \frac{\varepsilon}{p}, k) - \frac{1}{\lambda_1 - \frac{\varepsilon}{p}}\sum_k ||k||_\alpha^{-(\lambda_1 + \frac{\varepsilon}{q})-n}$$
$$= B(\lambda_1 - \frac{\varepsilon}{p}, \lambda_2 + \frac{\varepsilon}{p})\sum_k ||k||_\alpha^{-\varepsilon-n} - O(1)$$
$$> \frac{1}{\varepsilon}(\frac{\Gamma(\frac{1}{\alpha})}{\alpha^{n-1}\Gamma(\frac{n}{\alpha})}B(\lambda_1 - \frac{\varepsilon}{p}, \lambda_2 + \frac{\varepsilon}{p}) - \varepsilon O(1)).$$

Hence, by (20), (21) and the above results, we have the following inequality

$$\frac{\Gamma(\frac{1}{\alpha})}{\alpha^{n-1}\Gamma(\frac{n}{\alpha})}B(\lambda_1 - \frac{\varepsilon}{p}, \lambda_2 + \frac{\varepsilon}{p}) - \varepsilon O(1) < \varepsilon\widetilde{I} < \varepsilon M\widetilde{J} \leq M[\prod_{i=0}^{m-1}(\lambda_1 + i - \frac{\varepsilon}{p})]^{-1}(\frac{2^\varepsilon \Gamma(\frac{1}{\alpha})}{\alpha^{n-1}\Gamma(\frac{n}{\alpha})})^{\frac{1}{q}}. \tag{22}$$

For $\varepsilon \to 0^+$ in (22), in view of the continuity of the beta function, we find

$$\frac{\Gamma(\frac{1}{\alpha})}{\alpha^{n-1}\Gamma(\frac{n}{\alpha})}B(\lambda_1, \lambda_2) \leq M[\prod_{i=0}^{m-1}(\lambda_1 + i)]^{-1}(\frac{\Gamma(\frac{1}{\alpha})}{\alpha^{n-1}\Gamma(\frac{n}{\alpha})})^{\frac{1}{q}},$$

namely, $(\frac{\Gamma(\frac{1}{\alpha})}{\alpha^{n-1}\Gamma(\frac{n}{\alpha})})^{\frac{1}{p}}\prod_{i=0}^{m-1}(\lambda_1+i)B(\lambda_1,\lambda_2)\le M$. It follows that

$$M=(\frac{\Gamma(\frac{1}{\alpha})}{\alpha^{n-1}\Gamma(\frac{n}{\alpha})})^{\frac{1}{p}}\prod_{i=0}^{m-1}(\lambda_1+i)B(\lambda_1,\lambda_2)$$

is the best possible constant factor of (19).

The theorem is proved. □

**Remark 1.** *For* $\hat{\lambda}_1=\frac{\lambda-\lambda_2}{p}+\frac{\lambda_1}{q},\hat{\lambda}_2=\frac{\lambda-\lambda_1}{q}+\frac{\lambda_2}{p}=\lambda_2+\frac{\lambda-\lambda_1-\lambda_2}{q}$, *we find* $\hat{\lambda}_1+\hat{\lambda}_2=\lambda$,

$$0<\hat{\lambda}_1=\frac{\lambda-\lambda_2}{p}+\frac{\lambda_1}{q}<\frac{\lambda}{p}+\frac{\lambda}{q}=\lambda, 0<\hat{\lambda}_2=\lambda-\hat{\lambda}_1<\lambda.$$

If $\lambda-\lambda_1-\lambda_2\le q(n-\lambda_2)$, then we still can find $\hat{\lambda}_2\le n$. In the above case, we can rewrite (19) as follows:

$$\sum_k\int_0^\infty\frac{f(x)a_k}{(x+||k-\xi||_\alpha)^\lambda}dx<(\frac{\Gamma(\frac{1}{\alpha})}{\alpha^{n-1}\Gamma(\frac{n}{\alpha})})^{\frac{1}{p}}\prod_{i=0}^{m-1}(\hat{\lambda}_1+i)B(\hat{\lambda}_1,\hat{\lambda}_2)$$
$$\times[\int_0^\infty x^{p(1-m-\hat{\lambda}_1)-1}F_m^p(x)dx]^{\frac{1}{p}}[\sum_k||k-\xi||_\alpha^{q(n-\hat{\lambda}_2)-n}a_k^q]^{\frac{1}{q}}. \tag{23}$$

**Theorem 2.** *If* $\lambda-\lambda_1-\lambda_2\le q(n-\lambda_2)$, *the constant factor*

$$\prod_{i=0}^{m-1}(\lambda+i)(\frac{\Gamma(\frac{1}{\alpha})}{\alpha^{n-1}\Gamma(\frac{n}{\alpha})}B(\lambda_2,\lambda+m-\lambda_2))^{\frac{1}{p}}B^{\frac{1}{q}}(\lambda_1+m,\lambda\lambda_1)$$

*in (18) is the best possible, then we have* $\lambda-\lambda_1-\lambda_2=0$, *namely,* $\lambda_1+\lambda_2=\lambda$.

**Proof.** By Hölder's inequality (cf. [28]), we obtain

$$B(\hat{\lambda}_1+m,\hat{\lambda}_2)=\int_0^\infty\frac{u^{\hat{\lambda}_1+m-1}}{(1+u)^{\lambda+m}}du=\int_0^\infty\frac{1}{(1+u)^{\lambda+m}}u^{\frac{\lambda+m-\lambda_2}{p}+\frac{\lambda_1+m}{q}-1}du$$
$$=\int_0^\infty\frac{1}{(1+u)^{\lambda+m}}(u^{\frac{\lambda+m-\lambda_2-1}{p}})(u^{\frac{\lambda_1+m-1}{q}})du$$
$$\le[\int_0^\infty\frac{u^{\lambda+m-\lambda_2-1}}{(1+u)^{\lambda+m}}du]^{\frac{1}{p}}[\int_0^\infty\frac{u^{\lambda_1+m-1}}{(1+u)^{\lambda+m}}du]^{\frac{1}{q}}$$
$$=B^{\frac{1}{p}}(\lambda_2,\lambda+m-\lambda_2)B^{\frac{1}{q}}(\lambda_1+m,\lambda-\lambda_1). \tag{24}$$

In view of the assumption, compare with the constant factors in (18) and (23), we have the following inequality:

$$\prod_{i=0}^{m-1}(\lambda+i)(\frac{\Gamma(\frac{1}{\alpha})}{\alpha^{n-1}\Gamma(\frac{n}{\alpha})}B(\lambda_2,\lambda+m-\lambda_2))^{\frac{1}{p}}B^{\frac{1}{q}}(\lambda_1+m,\lambda-\lambda_1)$$
$$\le(\frac{\Gamma(\frac{1}{\alpha})}{\alpha^{n-1}\Gamma(\frac{n}{\alpha})})^{\frac{1}{p}}\prod_{i=0}^{m-1}(\hat{\lambda}_1+i)B(\hat{\lambda}_1,\hat{\lambda}_2)=(\frac{\Gamma(\frac{1}{\alpha})}{\alpha^{n-1}\Gamma(\frac{n}{\alpha})})^{\frac{1}{p}}\prod_{i=0}^{m-1}(\lambda+i)B(\hat{\lambda}_1+m,\hat{\lambda}_2),$$

namely, $B(\hat{\lambda}_1+m,\hat{\lambda}_2)\ge B^{\frac{1}{p}}(\lambda_2,\lambda+m-\lambda_2)B^{\frac{1}{q}}(\lambda_1+m,\lambda-\lambda_1)$; it follows that (24) retains the form of equality. We observe that (24) retains the form of equality if and only if there exist constants $A$ and $B$, such that they are not both zero and $Au^{\lambda+m-\lambda_2-1}=Bu^{\lambda_1+m-1}a.e.inR_+$ (cf. [28]). Assuming that $A\ne 0$, we have $u^{\lambda-\lambda_2-\lambda_1}=\frac{B}{A}a.e.inR_+$, namely, $\lambda-\lambda_1-\lambda_2=0$ and then $\lambda_1+\lambda_2=\lambda$.

The theorem is proved. □

## 4. Equivalent Forms and Operator Expressions

**Theorem 3.** *Inequality (18) is equivalent to the following inequality:*

$$
\begin{aligned}
J := \{ \sum_k ||k - \xi||_\alpha^{p\hat{\lambda}_2 - n} [\int_0^\infty \frac{f(x)}{(x + ||k - \xi||_\alpha)^\lambda} dx]^p \}^{\frac{1}{p}} \\
< \prod_{i=0}^{m-1} (\lambda + i) (\frac{\Gamma(\frac{1}{\alpha})}{\alpha^{n-1}\Gamma(\frac{n}{\alpha})} B(\lambda_2, \lambda + m - \lambda_2))^{\frac{1}{p}} B^{\frac{1}{q}} (\lambda_1 + m, \lambda - \lambda_1) \\
\times [\int_0^\infty x^{p(1-m-\hat{\lambda}_1)-1} F_m^p(x) dx]^{\frac{1}{p}}.
\end{aligned}
\tag{25}
$$

In particular, for $\lambda_1 + \lambda_2 = \lambda$, we reduce (25) to the equivalent form of (19) as follows:

$$
\begin{aligned}
\{ \sum_k ||k - \xi||_\alpha^{p\lambda_2 - n} [\int_0^\infty \frac{f(x)}{(x + ||k - \xi||_\alpha)^\lambda} dx]^p \}^{\frac{1}{p}} \\
< (\frac{\Gamma(\frac{1}{\alpha})}{\alpha^{n-1}\Gamma(\frac{n}{\alpha})})^{\frac{1}{p}} \prod_{i=0}^{m-1} (\lambda_1 + i) B(\lambda_1, \lambda_2) [\int_0^\infty x^{p(1-m-\lambda_1)-1} F_m^p(x) dx]^{\frac{1}{p}}
\end{aligned}
\tag{26}
$$

where the constant factor $(\frac{\Gamma(\frac{1}{\alpha})}{\alpha^{n-1}\Gamma(\frac{n}{\alpha})})^{\frac{1}{p}} \prod_{i=0}^{m-1} (\lambda_1 + i) B(\lambda_1, \lambda_2)$ is the best possible.

**Proof.** Suppose that (25) is valid. By Hölder's inequality (cf. [28]), we have

$$
I = \sum_k [||k - \xi||_\alpha^{\frac{-n}{p} + \hat{\lambda}_2} \int_0^\infty \frac{f(x)}{(x + ||k - \xi||_\alpha)^\lambda} dx][||k - \xi||_\alpha^{\frac{n}{p} - \hat{\lambda}_2} a_k] \leq J[\sum_k ||k - \xi||_\alpha^{q(n-\hat{\lambda}_2)-n} a_k^q]^{\frac{1}{q}}. \tag{27}
$$

Then by (25), we have (18).
On the other hand, assuming that (18) is valid, we set

$$
a_k := ||k - \xi||_\alpha^{p\hat{\lambda}_2 - n} [\int_0^\infty \frac{f(x)}{(x + ||k - \xi||_\alpha)^\lambda} dx]^{p-1}, k \in \mathrm{N}^n.
$$

If $J = 0$, then (25) is naturally valid; if $J = \infty$, then it is impossible to make (25) valid, namely $J < \infty$. Suppose that $0 < J < \infty$. By (18), we have

$$
\begin{aligned}
\sum_k ||k - \xi||_\alpha^{q(n-\hat{\lambda}_2)-n} a_k^q &= J^p = I \\
&< \prod_{i=0}^{m-1} (\lambda + i) (\frac{\Gamma(\frac{1}{\alpha})}{\alpha^{n-1}\Gamma(\frac{n}{\alpha})} B(\lambda_2, \lambda + m - \lambda_2))^{\frac{1}{p}} B^{\frac{1}{q}} (\lambda_1 + m, \lambda - \lambda_1) \\
&\quad \times [\int_0^\infty x^{p(1-m-\hat{\lambda}_1)-1} F_m^p(x) dx]^{\frac{1}{p}} [\sum_k ||k - \xi||_\alpha^{q(n-\hat{\lambda}_2)-n} a_k^q]^{\frac{1}{q}}, \\
\{ \sum_k ||k - \xi||_\alpha^{q(n-\hat{\lambda}_2)-n} a_k^q \}^{\frac{1}{p}} &= J \\
&< \prod_{i=0}^{m-1} (\lambda + i) (\frac{\Gamma(\frac{1}{\alpha})}{\alpha^{n-1}\Gamma(\frac{n}{\alpha})} B(\lambda_2, \lambda + m - \lambda_2))^{\frac{1}{p}} B^{\frac{1}{q}} (\lambda_1 + m, \lambda - \lambda_1) \\
&\quad \times [\int_0^\infty x^{p(1-m-\hat{\lambda}_1)-1} F_m^p(x) dx]^{\frac{1}{p}},
\end{aligned}
$$

namely, (25) follows, which is equivalent to (18).

The constant factor $(\frac{\Gamma(\frac{1}{\alpha})}{\alpha^{n-1}\Gamma(\frac{n}{\alpha})})^{\frac{1}{p}} \prod_{i=0}^{m-1} (\lambda_1 + i) B(\lambda_1, \lambda\lambda_2)$ in (26) is the best possible. Otherwise, by (27) (for $\lambda_1 + \lambda_2 = \lambda$), we would reach a contradiction that the constant factor in (19) is not the best possible.
The theorem is proved. $\square$

We set functions $\varphi(x) := x^{p(1-m-\hat{\lambda}_1)-1}$, $\psi(k) := ||k-\xi||_\alpha^{q(n-\hat{\lambda}_2)-n}$, then,

$$\psi^{1-p}(k) = ||k-\xi||_\alpha^{p\hat{\lambda}_2-n} \, (x \in R_+, k \in N^n).$$

Define the following real normed spaces:

$$L_{p,\varphi}(R_+) := \{f = f(x); ||f||_{p,\varphi} := (\int_0^\infty \varphi(x)|f(x)|^p dx)^{\frac{1}{p}} < \infty\},$$
$$l_{q,\psi} := \{a = \{a_{k_1,\cdots,k_n}\}; ||a||_{q,\psi} := (\sum_k \psi(k)|a_k|^q)^{\frac{1}{q}} < \infty\},$$
$$l_{p,\psi^{1-p}} := \{b = \{b_{k_1,\cdots,k_n}\}; ||b||_{q,\psi} := (\sum_k \psi^{1-p}(k)|b_k|^p)^{\frac{1}{p}} < \infty\},$$
$$\widetilde{L}(R_+) := \{f \in L_{p,\varphi}(R_+); f(x) = F_0(x) \geq 0, F_i(x) := \int_0^x F_{i-1}(t)dt \,(x \geq 0),$$
$$F_i(x) = o(e^{tx})\,(t > 0, i = 1, \cdots, m; x \to \infty)\}.$$

For any $f \in \widetilde{L}(R_+)$, setting $b_k := \int_0^\infty \frac{f(x)}{(x+||k-\xi||_\alpha)^\lambda}dx, k \in N^n$, we can rewrite (25) as follows:

$$||b||_{p,\psi^{1-p}} \leq \prod_{i=0}^{m-1} (\lambda+i)(\frac{\Gamma(\frac{1}{\alpha})}{\alpha^{n-1}\Gamma(\frac{n}{\alpha})}B(\lambda_2, \lambda+m-\lambda_2))^{\frac{1}{p}} B^{\frac{1}{q}}(\lambda_1+m, \lambda-\lambda_1)||F_m||_{p,\varphi} < \infty,$$

namely, $b \in l_{p,\psi^{1-p}}$.

**Definition 1.** *Define a Hilbert-type operator $T : \widetilde{L}(R_+) \to l_{p,\psi^{1-p}}$ as follows: For any $f \in \widetilde{L}(R_+)$, there exists a unique representation $Tf = b \in l_{p,\psi^{1-p}}$, satisfying $Tf(k) = b_k(k \in N^n)$. Define the formal inner product of $Tf$ and $a \in l_{q,\psi}$, and the norm of $T$ as follows:*

$$(Tf, a) := \sum_k a_k[\int_0^\infty \frac{f(x)}{(x+||k-\xi||_\alpha)^\lambda}dx] = I, ||T|| := \sup_{f(\neq 0) \in L_{p,\varphi}(R_+)} \frac{||Tf||_{p,\psi^{1-p}}}{||F_m||_{p,\varphi}}.$$

By Theorem 1, Theorem 2 and Theorem 3, we have

**Theorem 4.** *If $f \in \widetilde{L}(R_+), a \in l_{q,\psi}, ||F_m||_{p,\varphi}, ||a||_{q,\psi} > 0$, then we have the following equivalent inequalities:*

$$(Tf, a) < \prod_{i=0}^{m-1} (\lambda+i)(\frac{\Gamma(\frac{1}{\alpha})}{\alpha^{n-1}\Gamma(\frac{n}{\alpha})}B(\lambda_2, \lambda+m-\lambda_2))^{\frac{1}{p}} B^{\frac{1}{q}}(\lambda_1+m, \lambda-\lambda_1)||F_m||_{p,\varphi}||a||_{q,\psi}, \quad (28)$$

$$||Tf||_{p,\psi^{1-p}} < \prod_{i=0}^{m-1} (\lambda+i)(\frac{\Gamma(\frac{1}{\alpha})}{\alpha^{n-1}\Gamma(\frac{n}{\alpha})}B(\lambda_2, \lambda+m-\lambda_2))^{\frac{1}{p}} B^{\frac{1}{q}}(\lambda_1+m, \lambda-\lambda_1)||F_m||_{p,\varphi}. \quad (29)$$

Moreover, if $\lambda_1 + \lambda_2 = \lambda$, then the constant factor

$$\prod_{i=0}^{m-1} (\lambda+i)(\frac{\Gamma(\frac{1}{\alpha})}{\alpha^{n-1}\Gamma(\frac{n}{\alpha})}B(\lambda_2, \lambda+m-\lambda_2))^{\frac{1}{p}} B^{\frac{1}{q}}(\lambda_1+m, \lambda-\lambda_1)$$

in (28) and (29) is the best possible, namely, $||T|| = (\frac{\Gamma(\frac{1}{\alpha})}{\alpha^{n-1}\Gamma(\frac{n}{\alpha})})^{\frac{1}{p}} \prod_{i=0}^{m-1}(\lambda_1+i)B(\lambda_1, \lambda_2)$. On the other hand, if $\lambda - \lambda_1 - \lambda_2 \leq q(n-\lambda_2)$, the constant factor

$$\prod_{i=0}^{m-1} (\lambda+i)(\frac{\Gamma(\frac{1}{\alpha})}{\alpha^{n-1}\Gamma(\frac{n}{\alpha})}B(\lambda_2, \lambda+m-\lambda_2))^{\frac{1}{p}} B^{\frac{1}{q}}(\lambda_1+m, \lambda-\lambda_1)$$

in (28) or (29) is the best possible, then we have $\lambda - \lambda_1 - \lambda_2 = 0$, namely, $\lambda_1 + \lambda_2 = \lambda$.

**Remark 2.** *(i) For $\lambda = 1, \lambda_1 = \frac{1}{q}, \lambda_2 = \frac{1}{p}$ in (19) and (26), we have the following equivalent inequalities:*

$$
\sum_k \int_0^\infty \frac{f(x)a_k}{x+||k-\xi||_\alpha} dx < \prod_{i=0}^{m-1} \left(\frac{1}{q}+i\right)\left(\frac{\Gamma(\frac{1}{\alpha})}{\alpha^{n-1}\Gamma(\frac{n}{\alpha})}\right)^{\frac{1}{p}} \frac{\pi}{\sin(\pi/p)}
$$
$$
\times \left(\int_0^\infty x^{-pm} F_m^p(x)dx\right)^{\frac{1}{p}} \left[\sum_k ||k-\xi||_\alpha^{(q-1)(n-1)} a_k^q\right]^{\frac{1}{q}},
\tag{30}
$$

$$
\left[\sum_k |k-\xi||_\alpha^{1-n}\left(\int_0^\infty \frac{f(x)}{x+||k-\xi||_\alpha}dx\right)^p\right]^{\frac{1}{p}}
$$
$$
< \prod_{i=0}^{m-1} \left(\frac{1}{q}+i\right)\left(\frac{\Gamma(\frac{1}{\alpha})}{\alpha^{n-1}\Gamma(\frac{n}{\alpha})}\right)^{\frac{1}{p}} \frac{\pi}{\sin(\pi/p)}\left(\int_0^\infty x^{-pm}F_m^p(x)dx\right)^{\frac{1}{p}};
\tag{31}
$$

*(ii) for $\lambda = 1, \lambda_1 = \frac{1}{p}, \lambda_2 = \frac{1}{q}$ in (19) and (26), we have the following equivalent dual forms of (31) and (32):*

$$
\sum_k \int_0^\infty \frac{f(x)a_k}{x+||k-\xi||_\alpha} dx < \prod_{i=0}^{m-1} \left(\frac{1}{p}+i\right)\left(\frac{\Gamma(\frac{1}{\alpha})}{\alpha^{n-1}\Gamma(\frac{n}{\alpha})}\right)^{\frac{1}{p}} \frac{\pi}{\sin(\pi/p)}
$$
$$
\times \left[\int_0^\infty x^{p(1-m)-2} F_m^p(x)dx\right]^{\frac{1}{p}} \left[\sum_k ||k-\xi||_\alpha^{(q-1)n-1} a_k^q\right]^{\frac{1}{q}},
\tag{32}
$$

$$
\left[\sum_k ||k-\xi||_\alpha^{p-1-n}\left(\int_0^\infty \frac{f(x)}{x+||k-\xi||_\alpha}dx\right)^p\right]^{\frac{1}{p}}
$$
$$
< \prod_{i=0}^{m-1} \left(\frac{1}{p}+i\right)\left(\frac{\Gamma(\frac{1}{\alpha})}{\alpha^{n-1}\Gamma(\frac{n}{\alpha})}\right)^{\frac{1}{p}} \frac{\pi}{\sin(\pi/p)}\left[\int_0^\infty x^{p(1-m)-2}F_m^p(x)dx\right]^{\frac{1}{p}};
\tag{33}
$$

*(iii) for $p = q = 2$, both (30) and (32) reduce to*

$$
\sum_k \int_0^\infty \frac{f(x)a_k}{x+||k-\xi||_\alpha} dx < \frac{(2m-1)!!\pi}{2^m}\left(\frac{\Gamma(\frac{1}{\alpha})}{\alpha^{n-1}\Gamma(\frac{n}{\alpha})}\right)^{\frac{1}{2}}
$$
$$
\times \left[\int_0^\infty x^{-2m} F_m^2(x)dx\sum_k ||k-\xi||_\alpha^{n-1}a_k^2\right]^{\frac{1}{2}},
\tag{34}
$$

*and both (31) and (33) reduce to the equivalent form of (34) as follows:*

$$
\left[\sum_k ||k-\xi||_\alpha^{1-n}\left(\int_0^\infty \frac{f(x)}{x+||k-\xi||_\alpha}dx\right)^2\right]^{\frac{1}{2}}
$$
$$
< \frac{(2m-1)!!\pi}{2^m}\left(\frac{\Gamma(\frac{1}{\alpha})}{\alpha^{n-1}\Gamma(\frac{n}{\alpha})}\right)^{\frac{1}{2}}\left(\int_0^\infty x^{-2m}F_m^2(x)dx\right)^{\frac{1}{2}},
\tag{35}
$$

*The constant factors in the above particular inequalities (30)–(35) are all the best possible.*

**Remark 3.** *For $\alpha > 0$, we can only obtain $\frac{\partial}{y_j}h_x(y) < 0 (j = 1, \cdots, n)$ in (9). So, we cannot use Hermite–Hadamard's inequality to obtain (11) as well as other more accurate inequalities, but for $\xi = 0$, we still can obtain (11) by using the decreasingness property of series, and then the equivalent inequalities (18) and (25) for $\xi = 0$ with the best possible constant factor were proved.*

## 5. Conclusions

Hilbert-type inequalities with their applications played an important role in analysis. In this paper, following the way of [22], by using multi-techniques of real analysis, a more accurate half-discrete multidimensional Hilbert-type inequality with the homogeneous kernel as $\frac{1}{(x+||k-\xi||_\alpha)^\lambda}(x, \lambda > 0)$ involving one multiple upper limit function and the beta function is given in Theorem 1, which is a new extension of the published result in [22].

The equivalent conditions of the best possible constant factor related to several parameters are considered in Theorem 2. The equivalent forms, the operator expressions and some particular inequalities are obtained Theorem 3, Theorem 4 and Remark 2. The results are new applications of Hilbert-type inequalities involving multiple upper limit functions; the lemmas, as well as the theorems, provide an extensive account of these types of inequalities. The further study is to extend this paper's method to other types of Hilbert-type inequalities, for example, the Hilbert-type inequalities in whole plane.

**Author Contributions:** B.Y. carried out the mathematical studies, participated in the sequence alignment and drafted the manuscript. Y.H. and Y.Z. participated in the design of the study and performed the numerical analysis. All authors have read and agreed to the published version of the manuscript.

**Funding:** This work is supported by the National Natural Science Foundation of China (No. 62166011) and the Innovation Key Project of Guangxi Province (No. 222068071). We are grateful for this help.

**Data Availability Statement:** We declare that the data and material in this paper can be used publicly.

**Acknowledgments:** The authors thank the referee for his useful proposal to revise the paper.

**Conflicts of Interest:** The authors declare that they have no conflict of interest.

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
