# Peer review of "A More Accurate Half-Discrete Multidimensional Hilbert-Type Inequality Involving One Multiple Upper Limit Function"

_axioms, doi:10.3390/axioms12020211_

Round 1
Reviewer 1 Report
Report for Manuscript ID Axioms-2143067
In the paper entitled "A More Accurate Half-Discrete Multidimensional
Hilbert-type Inequality Involving One Multiple Upper Limit Function" submitted by Hong et al. using means of the weight functions and the idea of introduced parameters pertaining to the transfer formula and Hermite-Hadamard’s inequality, a more accurate half-discrete multidimensional Hilbert-type inequality with the homogeneous kernel as involving one multiple upper limit function is established. The equivalent conditions of the best possible constant factor related to several parameters are derived as well. Finally, the equivalent forms. the operator expressions and some particular inequalities are given.
To the best of my knowledge the results are new.
Anyway, the structure of the paper is not well-organized and also many typos and mistakes are found. The originality/novelty is average because there are many other published papers of this type from the last author.
The authors must show the significance of the paper and their advantages compared with other published papers of this type.
Moreover, they must include the application section to understand the importance. Furthermore, the conclusion section must be improved. I suggest adding some sentences regarding future research for interested readers.
The paper is not written using MDPI template.
If the above issues will be solved then I will recommend this paper for acceptance.
Sincerely yours,
Reviewer
Author Response
Dear Reviewer,
Please receive the attached files for reviewing.

Reviewer 2 Report
I wrote the comments and suggestions on the manuscript form at the report.

Author Response

(The authors gave the same response as above.)

Reviewer 3 Report
In this article, the authors have introduced and studied a new class of a more accurate half-discrete multidimensional Hilbert-type inequality with the homogeneous kernel involving one multiple upper limit function by means of the weight functions, the idea of introduced parameters, using the transfer formula and Hermite-Hadamard’s inequality. Also, the equivalent conditions of the best possible constant factor related to several parameters are considered. The equivalent forms. the operator expressions and some particular inequalities are obtained.
In my opinion, the paper is well organized, and the mathematical results of the paper are correct, interesting. But it can be accepted after considering the following corrections:
(1) Abstract: please elaborate a little bit.
(2) Introduction: please discuss a paragraph of the motivation of the present theoretical study supported by some recent references. Also, I expect a more complete description of what problem is being solved and why?
3) I find the literature needs a revision by inclusion of few more context specific recent works. Highlight the aim and novelty of your work.
(4) Remove the unnecessary equation numbers throughout the paper.
(5) The authors should put appropriate punctuation at the end of each equation throughout the paper.
(6) We prefer adding future work in the conclusion section.
(7) Are these new results sharp and more accurate compare with the others?
(8) By using a spell checker tool please check all the spelling of the paper.
(9) The authors should also check for any grammatical errors throughout the paper and advised to correct all the grammatical and formatting errors in the manuscript revised version, see
Page 2, "A point must be placed at the end of the equation 5". Alos, see the last equation on page number 7.
Page 3, Please clarify the meaning of the sum of G(k) in Lemma 2.
page 4, line no 2, Please write the form of the Hermite-Hadamard's inequality equation or write a reference to it.
page 8, please correct Holder’s throughout the paper.
page 11, line no 4, what do you mean Theorem 1-3.
(11) Is it possible to write the results obtained during this research in what is known as the time scale (please clarify).
(12) The literature of this paper can be enriched by adding the following related and recent literature:
a- "…" New Hilbert dynamic inequalities on time scales, Mathematical Inequalities & Applications, Volume 20, Number 4, (2017), 1017-1039.
With my best regards.
Author Response

(The authors gave the same response as above.)

Reviewer 4 Report
Thank you for inviting me to be a reviewer of the manuscript entitled A More Accurate Half-Discrete Multidimensional Hilbert-type Inequality Involving One Multiple Upper Limit Function. This document is really impressive in terms of your efforts to demonstrate the power of your study.
I recommend expanding the abstract of the study and supplementing the goals and contribution of the study.
I see great potential in the study for further follow-up research. Some passages of the study are confusing, repetitive and insufficiently clarified. I recommend expanding the conclusion of the study and adding a detailed discussion with references.
This study refers to 31 scientific references, resources and publications. The references used are not up-to-date and of insufficient quality and are therefore not a suitable theoretical basis for this study. Missing international publications. The reference list is poorly and inconsistently formatted. Some references are unnecessary and inappropriate. I recommend adding quality international literature and formatting the list of references appropriately.
This study represents a contribution in this area of research.
The basic ideas of the submitted manuscript are interesting.
Author Response

(The authors gave the same response as above.)

Round 2
Reviewer 1 Report
Report for the revised version of the paper entitled "A More Accurate Half-Discrete Multidimensional Hilbert-type Inequality Involving One Multiple Upper Limit Function" submitted by Hong et al.
The authors must show the significance of the paper and their advantages compared with other published papers of this type.
Moreover, they must include the application section to understand the importance.
If the above issues will be solved then I will recommend this paper for acceptance.
Sincerely yours,
Reviewer
Author Response
Please receive the attached file.

Reviewer 2 Report
In this peer review paper, authors prove some new versions of the using the transfer formula
and Hermite-Hadamard’s inequality, a more accurate half-discrete multidimensional Hilbert-type inequality. The results are correct.
must be check the first page has some correction.
Author Response
Please receive the attached file.

Reviewer 3 Report
Dear Editor:
I am pleased to inform you that the authors have made the required modifications and therefore I recommend the acceptance of the paper.
Author Response
Please receive the attached file.

Round 3
Reviewer 1 Report
Report for the revised version of the paper entitled "A More Accurate Half-Discrete Multidimensional Hilbert-type Inequality Involving One Multiple Upper Limit Function" submitted by Hong et al.
I accept after minor revision this second revised version.
Sincerely yours,
Reviewer